# Brain and Immune System: Intercellular Communication During Homeostasis and Neuroimmunomodulation upon Dysfunction

**DOI:** 10.3390/ijms26146552

**Published:** 2025-07-08

**Authors:** Volker Schirrmacher

**Affiliations:** Immune-Oncological Center Cologne (IOZK), D-50674 Cologne, Germany; v.schirrmacher@web.de

**Keywords:** dendritic cell, cytokine, extracellular vesicle, neuron, immunotherapy, synapse, T cell receptor

## Abstract

The review compares the principles of organization of the brain and immune system, two important organs developed over 500 million years in multicellular organisms, including humans. It summarizes the latest results from research in neurosciences and immunology concerning intercellular communication. While in the brain, intercellular communication is primarily based on exchange of electrical signals, this is not the case in the immune system. The question, therefore, arises as to whether nature developed two entirely different systems of organization. It will be demonstrated that a few basic principles of brain and immune responses are organized in a different way. A majority of intercellular communications, however, such as the formation of synapses, are shown to have many similarities. Both systems are intimately interconnected to protect the body from the1 dangers of the outside and the inside world. During homeostasis, all systems are in regulatory balance. A new hypothesis states that the central systems surrounded by bone, namely the central nervous system (CNS) and the central immune system (CIS), are based on three types of stem cells and function in an open but autonomous way. T cell immune responses to antigens from blood and cerebrospinal fluid protect the system and maintain neuroimmune homeostasis. The newly discovered tunneling nanotubes and extracellular vesicles are postulated to play an important role in crosstalk with already known homeostasis regulators and help in cellular repair and the recycling of biologic material. Three examples are selected to illustrate dysfunctions of homeostasis, namely migraine, multiple sclerosis, and brain cancer. The focus on these different conditions provides deep insights into such neurological and/or immunological malfunctions. Technological advances in neurosciences and immunology can enable neuroimmunomodulation and the development of new treatment possibilities.

## 1. Introduction

Learning by experience is a life-spanning process in both the brain and the immune system of multicellular organisms. Information about the external world is received via sensory organs, including the eye (visible light), nose (odors; chemical molecules), tongue (taste; chemical molecules), ear (sound waves), and skin (touch). Visual, olfactory, auditory, mechanosensory, thermosensory, or electrosensory nerve inputs are received via the sensory receptors of neurons and transmitted via sensory systems to the brain. In addition, the somatosensory system, spread throughout the body, surveys the inside world and is regulated by the receptors of respective neurons [1,2]. Information about the internal world of the body is also received and processed via cells from the innate and adaptive immunity system [3,4]. It is now established that both systems are interconnected in a variety of ways and have a life-protecting function against dangers from the outside and inside worlds.

This overview sheds light on the main principles as to how the brain and the immune system work. It focuses on intercellular communication, with special focus on cognate interactions through synapses. It also elucidates similarities and differences between the two systems. In a healthy body, intercellular communication via neuroimmune interfaces and networks has been shown to ensure homeostasis in the brain and the immune system. Examples of neuroimmune interface dysfunction include migraines (worldwide prevalence 15%), multiple sclerosis (MS) (the most frequent autoimmune disease of the central nervous system, CNS), and brain cancer (the most frequent primary malignant brain tumor). Apart from their relevance, these diseases were selected as examples for meningeal lymphatic vessel (MLV) dysfunction, neuroimmune dysfunction, and neuro-glia dysfunction. Concepts and examples of the neuroimmunomodulation of these illnesses are described. With regard to brain cancer, it is proposed, based on positive results from clinical studies, to develop new combination treatments, including immunotherapy.

The criteria for literature selection were based on topic relevant keywords and searches in the Pubmed library. Selection criteria were relevance and novelty. The profile of the review differs from other timely publications about the brain and immune system. While the latter report on distinct selected aspects, this review provides a broad comparative overview of both systems. In addition, it includes a new hypothesis about the homeostasis of the brain and immune system in which the brain, bone marrow, and three types of stem cells play an important role.

## 2. Intercellular Communication

This chapter provides some basic facts from textbooks about the brain [1,2] and the immune system [3,4] before presenting some of the latest findings from these different fields of research.

### 2.1. The Brain

#### 2.1.1. Basic Information

The brain is a very active organ with a constant high demand of nutrients and oxygen to be converted into energy. In neurons, mitochondria are the primary suppliers of energy. An electrochemical proton gradient between the peri-mitochondrial cleft and the inner mitochondrial membrane drives the production of adenosine triphosphate (ATP) from ADP and Pi, catalyzed by the ATP synthase. The mitochondrial production of ATP is coupled to the respiration chain oxidizing the major products of glycolysis from the cytosol, namely pyruvate and nicotinamide-adenin-dinucleotide (NADH). The mitochondrial proton turbines pump and produce 150 ATP reactions and molecules per second, operating with astounding efficiency [5].

The CNS, derived from neuroectoderm, is composed of three parts, namely the cerebrum, the cerebellum, and the brainstem with its spinal cord. Intercellular communication between these parts is organized through neuronal networks [1,2].

A neuron has a signal receptor segment (a pericaryon with dendrites), a signal transition element (an axon), and a signal transfer element (the presynaptic terminal). Neurons build synaptic contacts with other neurons or with nonneuronal cells (e.g., from skeletal muscle fibers or endocrine glands) for information transfer. Neuron function includes ion movement and much electrophysiology, including membrane potentials, postsynaptic potentials, action potentials, and voltage clamps [1,2].

#### 2.1.2. Neuronal Synapses

The brain is a complex organ comprising neurons, glia (astrocytes, oligodendrocytes, and microglial cells), and more than 1 × 10^14^ synapses [6]. One neuron can make up to ten thousands synapses. Cerebral cortex neurons are arranged in specific layers and form connections both within the cortex and with other brain regions [7]. Glial cells are found from invertebrates to humans with morphological specializations related to the neural circuits in which they are embedded [6]. They modulate neuronal functions, brain wiring and axon myelination, dendritic spine structure, and information processing [8]. Astrocytes can send processes to the synaptic cleft and communicate with pre- and postsynaptic neurons in a tripartite way [7,8]. In addition to glial cells, the brain is also surveyed by classical immune cells, although at a lower number than in other organs.

Chemical signals are transmitted between neurons via synapses (20–40 nm distance) releasing neurotransmitters. A well-coordinated group of presynaptic proteins mediate the synaptic neurotransmitter release. Upon the arrival of an action potential, they control and trigger fusion between synaptic vesicles and the presynaptic neuron terminal [9]. Neurotransmitters exert a neuroregulatory and also an immunoregulatory role [10].

#### 2.1.3. Neuronal Network Visualization

The first magnetic resonance image (MRI) of the human body was obtained in 1974 and had a contrast quality much better than images obtained by scanner radiodensity tomography. A further improvement has been achieved through diffusion MRI (fMRI). Diffusion-weighted MRI and diffusion tensor imaging are being used to map white matter tractography (a 3D modeling technique) in the living human brain. These sophisticated MRI techniques allow researchers to visualize the brain’s extraordinarily complex neuronic fiber bundles and their anatomic connections in health and disease situations. Each nerve contains fibers arranged in parallel within strands enveloped by connective tissue. Structure–function analyses are performed by creating so called connectoms. Macroscale (blood flow; functional) and microscale (water diffusivity; structural) connectoms (neuronal wiring diagrams) have created tens of large-scale datasets, e.g., from the cortex, cerebellum, retina, peripheral nervous system (PNS), and neuromuscular junctions [11].

#### 2.1.4. CNS Versus PNS

The PNS is divided into an autonomous and a somatic branch. The former is further divided into sympathicus and parasympathicus branches, while the latter is divided further into a motor output branch and a sensory input branch.

In the autonomous PNS branch, the brain sends efferent commands (neural or hormonal) to peripheral target tissues through peripheral autonomic pathways. Distinct neural circuits in the spinal cord, brainstem, and hypothalamus connect to peripheral autonomic pathways and to the organs of the head, as well as to the heart, lungs, gastrointestinal tract, skin, skeletal muscle joints, and pelvic organs. Afferent signals from the periphery to the brain are neural, hormonal, or physicochemical (e.g., blood glucose level). Spinal reflex circuits are integrated in homeostatic regulations represented supraspinally in the brainstem and hypothalamus [1,2].

#### 2.1.5. Neuronal Intercellular Signal Communication

Neurons are electrically and chemically excitable. In most resting neurons, the membrane potential is between −60 and −75 millivolts. This means that the neuron is polarized: the inside of the plasma membrane has a negative charge compared to the outside. Opening and closing ion channels alters the membrane potential. Neurons communicate electrical and chemical signals across synapses. In the electrical synapses of neurites (axons or dendrites) the action potential is transferred directly to the adjacent cell without the help of neurotransmitters. Often such synapses use membrane channels called gap junctions. Such gap junctions (about 3.5 nm across) mediate the transmission of electrical impulses, calcium fluxes, and small molecules. Calcium ions have an important signaling function in the cytoplasm [12] and in mitochondria [5] for cellular energy supply and metabolism [12]. In the last decade, astrocytes have emerged as a critical entity within the brain because of their unique role in recycling neurotransmitters, actively modulating the ionic environment and regulating metabolism of cholesterol and sphingolipids, important components of lipid rafts [13]. Thus, astrocytes enable the fine-tuning of synaptic receptors that control the flow of glutamate in the tripartite synapse. Gangliosides can generate an active electrical field for cationic neurotransmitters, such as serotonin. Conversely, they can generate a repulsive electric field for anionic neurotransmitters, such as glutamate [14]. Common neurotransmitters include chemicals, such as serotonin, epinephrine, dopamine, acetylcholine, and gamma-aminobutyric acid. They transmit excitatory or inhibitory signals to target cells carrying respective cell surface receptors. Neurotransmitter action can involve ionotropic receptors (ligand-gated ion channels) or G-protein-coupled receptors [1,2]. To evaluate effects of neurochemicals, drugs, and toxins, strategies for electrochemical biosensing analysis have been developed and reviewed [15]. Another recent review concerns the process from neurotransmitters to networks [16]. Neuronal networks are organized across micro-, meso-, and macro-scale principles and distances [16].

#### 2.1.6. New Discoveries

Neurons are highly polarized cells that are able to compartmentalize biological processes in space and time via the parallel processing of many reactions [17]. Recent studies revealed that synaptic contacts use different protein combinations. These appear to define the specificity, function, and plasticity potential of synapses. The studies involved the microdissection of 5 different brain regions with fluorescent-activated synaptosome sorting from 7 different transgenic mouse lines to analyze the proteomes of 18 different synapse types. About 1800 unique synapse-type-enriched proteins were identified in addition to shared synaptic protein modules. The latter highlighted endocytosis and vATPases (ancient vacuolar-type ATPase enzymes) as generic synaptic protein modules. These are used by synapses independent of neurotransmitter type. In contrast, the exocytosis machinery, the presynaptic active zone, transsynaptic adhesion molecules, and postsynaptic protein modules make use of different sets of proteins for different synapse types [17].

Correlation network analysis revealed excitatory and inhibitory protein communities. It was concluded that the discovered plasticity and synaptic diversity provides a molecular basis for learning and memory formation [17,18].

Another recent study investigated molecular and cellular dynamics of the developing neocortex at single-cell resolution [19]. It included 38 human neocortical samples encompassing the prefrontal cortex and the primary visual cortex. The study elucidated intricate cell–cell communication networks during development, emphasizing robust interactions between glutamatergic excitatory neuron (EN) and GABAergic inhibitory neuron (IN) subclasses [19]. Interestingly, the study identified Tri-IPC, a tripotential intermediate progenitor subtype cell which is responsible for local production of GABAergic neurons, oligodendrocyte precursor cells, and astrocytes. Remarkably, glioblastoma multiforme (GBM) cells were found to resemble Tri-IPCs at the transcriptomic level. From this finding, it was suggested that cancer cells might hijack developmental processes to enhance growth and heterogeneity [19].

Multi-protein complexes enriched in K-CL co-transporter 2 (Kcc2) from brain plasma membranes have been shown to contain a set of novel structural ion transporting immune related and signaling protein interactions. They are present at both excitatory and inhibitory synapses. Furthermore, 246 Kcc2 associated proteins were identified, representing components of the neuronal cytoskeleton and signaling molecules that may regulate transporter membrane trafficking, stability, and activity [20].

#### 2.1.7. Intercellular Communication in the Brain by TNTs and EVs

Tunneling nanotubes (TNTs) are recently recognized structures for intercellular communication. TNTs differ from tumor microtubes in size and life span [21]. They are F-actin-containing thin protrusions of the plasma membrane of a cell and allow direct physical connection to the plasma membrane of remote cells. TNTs allow cells to connect a long distance and facilitate the exchange of membrane components and cytoplasmic material, including mitochondria [22]. Astrocyte-to-neuron transportation of proteins in the cerebral cortex was described to require F-actin dependent TNTs [23]. TNTs may play a role in normal homeostatic processes in the eye as well as in ocular disease [24]. A recent study describes the microglial rescue of neurons from aggregate-induced dysfunction and death through TNTs [25]. However, TNT transfer can also mediate pathological conditions (e.g., Creutzfeldt–Jakob disease). Prions have been reported to hijack TNTs for intercellular spread in the CNS [26].

In the last fifteen years, it has been demonstrated that both neurons and glial cells release extracellular vesicles (EVs) [27]. EVs play an important role in intercellular as well as interorgan communication. EVs also communicate with the complement system in neurological diseases [28], such as MS (see Section 4.6).

Adipose tissue-derived EVs exert metabolic actions in distal organs, such as the liver, skeletal muscle, pancreas, and brain [29]. EVs from immune cells contain molecules capable of activating molecular pathways that aggravate neuroinflammatory processes in neurological disorders [30]. EVs can cross the blood–brain barrier so that peripheral immune signals can influence brain function via EV -mediated communication. Interestingly, EVs from mesenchymal stem cells (MSCs) have the potential to reduce neuroinflammation and cognitive deficits [29,30]. MSCs also have the capacity to form TNTs and transfer mitochondria to target cells, leading to changes in cell energy metabolism and functions [31].

Therapeutic potential has been ascribed to apoptotic vesicles in modulating inflammation, immune responses, and tissue regeneration [32]. These vesicles can induce apoptosis in cancer cells, provide tumor antigens for cancer vaccines (see Section 4.7), and even serve as drug delivery systems [32].

EVs can be categorized as exosomes, ectosomes, apoptotic bodies, large oncosomes and migrasomes.

#### 2.1.8. Summary

In summary, intercellular communication in the brain occurs via electrical and chemical signals in a variety of ways, including (i) direct cell contact channels (gap junctions) for the transmission of electrical impulses and for the exchange of small molecules, (ii) interneuronal synapses for the electrical impulse-mediated release of neurotransmitters, and (iii) TNTs and EVs from neurons, astrocytes, immune cells, and MSCs for short-, medium-, and long-distance communication and exchange of biologics.

### 2.2. The Immune System

Before describing neuroimmune interfaces, we focus on the immune system itself and reflect upon its development and function.

#### 2.2.1. Basic Information

The adaptive immunity system of vertebrates, derived from the mesoderm, developed about 500 million years ago [33]. The yolk sac and fetal liver are the fetal hematopoietic organs and the post-natal bone marrow (BM) is the organ where hematopoietic stem cells (HSC) and MSCs form the basis for hematopoiesis, osteogenesis, and immune responses [34]. HSCs in the BM give rise to common myeloid (CMP), common lymphoid (CLP), and common basophil–mast cell (BMCP) progenitors that differentiate into the mature leukocytes that populate the body [3,4].

#### 2.2.2. Cognate Antigen Recognition by B and T Lymphocytes

Lymphocytes are the only cells capable of specifically recognizing antigens and are, thus, the principal cells of adaptive immunity. They can be compared to neurons, which are the principal cells of the brain. The adaptive immune response is supported by innate immunity cells, such as phagocytes (neutrophils, monocytes, and macrophages), mast cells, basophils, eosinophils, dendritic cells (DCs), and natural killer (NK) cells [3,4].

The antigen receptors of B and T cells are encoded by a limited number of gene segments (BCR: Ig light chain kappa, Ig light chain lamda, and Ig heavy chain, TCR: alpha, beta, gamma, and delta chains) that are spatially segregated in the germline loci but somatically recombined in CLP cells, leading to the development of B and T cells.

The total potential repertoire of BCRs is about 10^11^, that of α and β T cells is about 10^16^, and that of γ and δ T cells about 10^18^ [2,3]. While B cells, with their antibody-like receptors, can recognize antigens directly, T lymphocytes require that antigens are first processed and then presented by professional antigen-presenting cells (APCs). The most important APCs for T cells are DCs. These cells have dendrites, like neurons, and sample tissues for the presence of self or non-self-antigens. In the case of self-antigens (SAs) they transmit a signal of non-action (tolerance) to T cells [3,4], while in the case of non-self-antigens (NSAs, neoantigens), they eventually transmit a signal of action [34].

#### 2.2.3. Immune Regulation for the Prevention of Autoimmune Reactivity

To avoid autoimmune reactivity, the immune system has evolved various control checks. A three signal model of naïve CD4 T helper (Th) cell activation and naïve CD8 T killer precursor (Tc) cell activation has been proposed [4]. The control checks consist of the dependency of TCR-mediated T cell activation (signal 1) on a costimulatory CD28 mediated signal (signal 2). Autocrine cytokines (e.g., IL-2) deliver signal 3 to activated Th cells via CD25 while paracrine cytokines (e.g., IL-2) from the microenvironment deliver signal 3 to activated Tc [2,3].

Peripheral tolerance is also regulated by T regulatory (Treg) cells, which deliver T cell inhibitory signals mediated by regulatory cytokines, such as IL-10 and transforming growth factor ß (TGFß) [2,3].

A further control consists of T cell inhibitory receptors (e.g., CTLA-4, PD-1, LAG-3, and Tim-3). The discovery of cancer therapy via the inhibition of negative immune regulation (e.g., via immune checkpoint inhibitory antibodies) was awarded a Nobel Prize in Medicine in 2018.

#### 2.2.4. Antigen Presentation to T Cells by Dendritic Cells

Like in the brain, synaptic cellular interactions play a decisive role in the adaptive immunity system. DCs distributed throughout the body constantly take samples (via macropinocytosis) from their environment and present peptides from digested proteins via major histocompatibility (MHC) class I (MHC I) or class II (MHC II) molecules at their surface to T cells which circulate through blood and the lymphatic system. CD8 T cells can recognize peptide–MHC I (pMHC I) complexes on all nucleated cells of the body. CD4 T cells recognize pMHC II complexes primary expressed by APCs [3,4]. The lymph nodes, spleen, and BM provide optimal local conditions for facilitating the rare event of cognate T-APC interactions. Circulatory T cells entering those lymphoid organs through blood vessels (via transendothelial migration) patrol the respective parenchymal environment in search of APCs. Upon APC scanning, they discover SAs or NSAs presented at their cell surface. In the case of SAs, the T cells receive a tolerance signal and leave the APC after a short contact [34].

#### 2.2.5. Immunogenic T-APC Interaction

In the case of recognized NSAs (neoantigens) and costimulatory and danger signals, a cognate T-APC interaction event takes place. This leads to the formation of an immunological synapse [3,4,34,35]. The two types of cells of the synapse, namely T cells and DCs, stay together and exchange signals that are important for later memory T cell (MTC) formation [34].

Such a T cell mediated immune response goes through several programmed phases of immune cell interactions, including (i) APC scanning, (ii) APC-T cluster formation, (iii) the generation of T lymphoblasts, (iv) clonal T cell expansion within such clusters, (v) differentiation to effector T cells, and (vi) the release of activated, expanded, and differentiated T cells from the cluster. Within 10 days, NSA-specific T cell immune responses lead to effector T cells, with CD4 helper T cells releasing cytokines or cytolytic CD8 T cells killing NSA-expressing target cells [34,35].

#### 2.2.6. Formation of an Immunological Synapse

Several molecular interactions are required for T cells to become activated by APCs, namely (i) TCR-pMHC, (ii) CD4/8-MHC, (iii) LFA-1-ICAM-1, and (iv) CD28-CD80 [36]. These lead to the folding of an immunological synapse [37], with supramolecular activation complexes (SMACs) at the periphery (p-SMAC) and the center (c-SMAC).

#### 2.2.7. Cytoskeletal Reorganization and T Cell Polarization

Physical T cell–APC contact is a prerequisite for juxtacrine immunological synapse signaling. It sets up an axis for polarization of TCR, adhesion molecules, kinases, cytoskeletal elements and organelles [38,39,40]. The immunological synapse directs the polarized transport and communication across the synaptic cleft [41]. It also mediates the generation and exocytic release of bioactive microvessels that have been termed synaptic ectosomes [42]. Ectosomes export the TCR, linked to the functional CD40 ligand, from Th cells [42]. Sorting nexin 27 (SNX27) enables translocation of the microtubule organizing center (MTOC) and of the secretory machinery to the immune synapse [43].

A naïve CD4 Th cell activated by an APC (DC or B cell) proliferates and polarizes in the presence of differentiation factors into distinct subsets, such as Th1, Th2, or Th17 cells [3]. The cell fate decision is influenced by two general mechanisms of regulation of gene expression, namely micro-RNA and epigenetic modification. Th1 cells combat intracellular pathogens and Th2 cells combat extracellular pathogens. Th17 cells are inflammatory, associated with autoimmune disease (see for instance MS, Section 4.6) and combat bacteria and fungi. Th2 cells are associated with allergy and Th1 cells with transplant and tumor rejection [3,4] (see GBM, Section 4.7).

#### 2.2.8. Priming of Mitochondria of Importance for MTC Differentiation

Costimulatory signals via CD28 prime mitochondria [43] to remodel cristae, develop spare respiratory capacity, facilitate fatty acid oxidation, and rapidly produce cytokines upon restimulation. Without CD28, TCR signaling can elicit primary effector T cells but not MTCs [44].

The life span of MTCs can be up to 50 years, while that of naïve T cells is 5–7 weeks and that of effector T cells (Th or CTL) is 2–3 days [3,4].

#### 2.2.9. Tripartite T-APC-T Cell Interactions

In addition to T-APC interactions and T-B cell cooperation, cellular interactions also exist between CD4 and CD8 T cells and there are even tripartite interactions between APCs, CD4, and CD8 T cells. CD4 T cell help is required for CD8 T cell memory and involves CD25 [45,46].

#### 2.2.10. Intercellular Communication via Cytokines

Cytokines are of great importance in the immune system for intercellular communication. Major human cytokines and cytokine receptors include interferon α, β, and γ, interleukins (1–33 types), TNF-related cytokines (3 types), and transforming growth factors (7 types) [3,4]. Cytokines important for the differentiation of Th1 cells are IL-2, IL-12, IFN-γ, and IL-27. Th2 cells require IL-4 and Th17 cells, as well as TGFß, IL-6, and IL-21 for differentiation [4]. Distinguishing surface markers include IL-12R (Th1), IFNγR (Th2), and IL-23R (Th17).

#### 2.2.11. The CIS and PIS

Like the nervous system with its CNS and PNS parts, the immune system can be subdivided into the central immune system (CIS) and the peripheral immune system (PIS). Central tolerance (thymus, BM) and peripheral tolerance (peripheral tissues) to self-antigens are two examples of this division. Antigens may be administered in ways that induce tolerance rather than immunity. When applied into the periphery, adjuvants are normally necessary to induce immune responses. This method of antigen application can be exploited for the prevention and treatment of transplant rejection and autoimmune and allergic diseases [3,4].

#### 2.2.12. Intercellular Communication by TNTs and EVs

TNTs, as described in the section discussing the brain, also occur between cells of the immune system. For instance, macrophage polarization to either proinflammatory M1 type cells releasing IFN-γ or anti-inflammatory M2 type cells releasing IL-4 and IL-10 was reported to have an impact on TNT formation and the intercellular trafficking of organelles (lysosomes and mitochondria) [47]. Another example relates to cell–cell connections between human antigen-presenting B and T lymphocytes [48]. When DCs and monocytes were triggered to flux calcium by chemical or mechanical stimulation, the signal could be propagated via TNTs within seconds to other cells at distances hundreds of microns away [49]. Nonneuronal cells can, thus, transmit signals, like neuronal cells, to distant cells through a physically connected network.

Like TNTs, EVs also contribute to intercellular communication within the immune system. They can be used for cancer immunotherapy [50] (see Section 4.7). T cell messages can be transmitted by extracellular vesicles from T cell microvilli [51].

#### 2.2.13. Similarities Between the Brain and the Immune System

There are similarities in the developmental and organizational principles between the brain and the immune system. The tripartite precursor cell Tri-IPC can be compared to the common lymphoid precursor cell CLP. While Tri-IPCs lead to development of neurons, oligodendrocyte precursors, and astrocytes, CLPs lead to the development of lymphocytes, myeloid precursors, and mast cells. Synapses in both systems include tripartite cellular interactions. SNX27 recycles intracellular cargo in immune and neuronal synapses. Defective expression is associated with a failure in cytoskeleton organization in T cells and with neuronal synapse dysfunction, cognitive impairment, anxiety, and stress susceptibility [43,52].

In the brain, synapses are composed of a presynaptic neuron, postsynaptic neuron, and astrocyte, while in the adaptive immunity system, they are composed of a CD4 T cell, CD8 T cell, and DC as an APC. Neurons function via excitatory and inhibitory protein communities. T cells function via positive and negative signaling receptors.

#### 2.2.14. Summary

In summary, the principles of intercellular communication in the immune system show differences and similarities to those of the brain. (i) Communication via electrical impulses is restricted to neurons and the neuronal network, (ii) communication between gap junctions occurs in both systems, (iii) communication via synapses holds true for both systems, (iv) somatic recombination of TCRs and BCRs and the distinction between self- and non-self-antigens are specialties of the immune system, (v) T-APC immune synapses communicate exclusively via chemical signals, while interneuronal synapses combine electrical and chemical signals, (vi) communication via TNTs and EVs occurs in both systems, (vii) and communication via cytokines plays an important role in the immune system.

Typical features of Section 2 are summarized in Table 1.

## 3. Neuroimmune Interfaces and Network Communication

### 3.1. Basic Information

The neuroimmune network consists of immune cells and immune-derived molecules, endocrine glands and hormones, the nervous system, and neuro-derived molecules. The nervous system regulates the function of immune cells through neurotransmitters or neuropeptides [1,2]. Conversely, immune cells survey the nervous system via gliotransmitters [12,13] and play a key role in neuronal injury repair [3,4]. The neuroimmune network is tri-directional, with communication between lymphocytes, neurons, and endocrine gland cells.

How the immune system shapes nervous system function has been the topic of a recent review with a focus on cytokines and cytokine signaling in neurons [53]. For example, GABAergic IL-4 receptors expressed on neurons mediate T cell effects on synaptic function and episodic memory [54]. The effects of the nervous system on immunity have also been reviewed with respect to why, how, and where they occur [55]. Neuronal–immune crosstalk involves neurotransmitters (e.g., calcitonin gene-related peptide (CGRP)) or neuropeptides and respective receptors on immune cells [55].

Immune dynamics in the CNS and its borders have been reviewed with a focus on homeostasis and disease. It was concluded that CNS border sites are inhabited by a plethora of different innate and adaptive immune cells in contrast to the microglia, which are largely responsible for surveying brain parenchyma [56].

Anatomical sites for neuroimmune interactions at brain borders include skull bone marrow, perivascular spaces, meninges, the choroid plexus, and the subarachnoid lymphatic-like membrane (SLYM) [57].

### 3.2. Communication via SLYM

The CNS is lined by meninges—dura, arachnoid, and pia mater. A fourth meningeal layer, the subarachnoid lymphatic-like membrane (SLYM), has been described, which encases blood vessels and immune cells [58]. SLYM permits the direct exchange of small solutes between cerebrospinal fluid (CSF) and venous blood [58]. The inner skull of the cortex contains vascular channels which provide a passageway between the skull and the brain parenchyma for cells and CSF-derived antigens and hormones [59]. Dysfunction of this glymphatic drainage system might be involved in migraines (see Section 4.5) and various other neurological disorders [60].

Initially described as a protective sac to cover the brain, the meninges are now recognized as an active participant in diverse brain functions. Meningeal-derived cytokines, chemokines, and growth factors contribute to brain development, neuronal firing, neuronal connectivity, and homeostatic rodent behaviors. Specialized vasculature in the meninges enables continuous T cell trafficking and interactions with resident APCs. CSF has access to the different meningeal layers, enabling T cell surveillance of brain-derived antigens.

### 3.3. Communication via Gateway-Specific Blood Vessels

Novel neuroimmune communications at specific vessels are described by gateway reflexes [61]. In these reflexes, immune cells bypass the blood–brain barrier and infiltrate the CNS, thereby causing neuroinflammation. Examples include specific blood vessels in the spinal cords and brain in experimental autoimmune encephalomyelitis and systemic lupus erythematosus models, retinal blood vessels in an experimental autoimmune uveitis model, and the ankle joints in an inflammatory arthritis model [61].

Several environmental stimuli, for instance gravity, pain, psychological stress, and light, can activate specific neural pathways and establish immune cell gateways at specific blood vessels, including the CNS, causing tissue-specific inflammatory diseases.

(i)Gravity gateway reflex: Gravity activates sensory nerves in the soleus muscle, whose cell bodies are located at the dorsal root ganglion of the fifth lumbar spinal cord (L5). Soleus muscles are anti-gravity muscles and are necessary to maintain posture for weight-bearing. The gateway reflex is created via the sympathetic pathway from L5 sympathetic ganglions. Characteristic mediators at the L5 vertebra are autoreactive CD4 T cells, norepinephrine (NE), and chemokine CCL20, an IL-6 amplifier [61]. CCL20 attracts CCR6-expressing immune cells in an EAE mouse model, such as Th17 cells, which are also involved in MS (see Section 4.6).(ii)Pain-induced gateway reflex: Pain stimuli are delivered to the anterior cingulate cortex (ACC), which has neurons related to pain sensation. This activation finally reaches the L5 vertebrate via sympathetic nerves to induce the release of CX3CL1 from ventral blood vessels. Characteristic components of the reflex include autoreactive CD4 T cells, NE, CX3CR1 + monocytes, and CX3CL1 chemokines [61].(iii)Stress gateway reflex: Chronic mental stresses sequentially activate neurons in the paraventricular nucleus of the hypothalamus and other neurons connecting specific vessels adjacent to the third ventricle, dentate gyrus, and thalamus. This neural activation induces the CCL5-expression-dependent accumulation of CD4 T cells and MHC class II**^hi^** monocytes. The induced microinflammation in specific brain vessels activates the dorsomedial nucleus of hypothalamus, propagates activation signals to dorsal motor nucleus of the vagus nerve, and finally causes upper severe gastrointestinal (GI) tract failure. In the investigated mouse model, the stomach epithelial cell damage induced bleeding, and the acute elevation of cytosolic potassium ions caused sudden cardiac dysfunction and death [61].(iv)Light gateway reflex: This reflex [62] is novel in that it negatively regulates injured vascular endothelial cells to provide a protective effect in the retina of experimental autoimmune uveitis (EAU) mice. The renal inflammation by photoreceptor peptide-specific CD4 T cells could be reduced by photopic (visual) light, which simulates strongly neurons in retina tissue [62].

### 3.4. Communication Between the Bone Marrow and Brain

Interconnection and communication between BM—the central immune system (CIS) [34]—and the CNS have recently been described [63]. Three-dimensional micro-anatomic investigations of BM revealed adrenergic nerves running adjacent to arteries and arterioles [64]. In another study, adrenergic nerve fiber terminals from the sympathicus were visualized to end at microtubular and microfilament walls in the cytoplasm of a subset of DCs [65]. While adrenergic nerve endings can exert activating signals, cholinergic nerve endings from the parasymphaticus can exert a dampening effect on immune cells in the BM. A neuro-osteogenic network was described as being of importance for the tissue engineering of bone regeneration [66]. The ways in which the PNS communicates with bone-lineage cells, the vasculature, and immune cells in the bone microenvironment have been described [67].

The BM as a priming site for T cell responses to bloodborne antigens was first described in 2003 [35]. Mature circulating naïve T cells can home in on BM sinuses after they have passed through aortic arteries and arterioles. There, they transmigrate through the sinus endothelium and enter the parenchyma. This contains resident DCs with the capacity to function as APCs. The T cell response in BM is autonomous and independent from secondary lymphoid organs (lymph nodes and spleen) [35]. Thus, BM is an active antigen-responsive organ.

BM is also a very active cell-generating organ, constantly providing blood cells and bone cells in finely tuned homeostasis [34]. The spongy architecture of BM provides niches for cells of great importance for homeostasis, health, and protection against pathogens, such as stem cells and immune memory cells, including plasma cells and MTCs [34].

A large pool of BM is situated in the skull and vertebrae (see Figure 1). Under homeostasis, immune cells produced in the skull BM can utilize channels to travel into the dura mater. Under pathological conditions, these cells can further traffic to underlying leptomeninges and to brain and spinal cord parenchyma. This system represents a cerebrospinal fluid (CSF)-to-skull BM axis so that this local BM can respond to CNS inflammation [61].

In situ two-photon microscopy revealed cross-presentation of bloodborne antigens to naïve T cells in the BM of the skull [68], in common with the findings from the vertical column and pelvis [35] (see Figure 1). Naïve CD8+ T cells moved rapidly under steady state conditions but arrested their motion immediately upon sensing antigenic peptides. Antigen-specific T cells decelerated, clustered, and upregulated CD69, and were observed dividing in situ to yield effector cells [68]. A small proportion of such specific T cells from the BM of skull develop into MTCs and enter into MTC specific niches of skull BM. Such MTCs provide superior immune surveillance and protective functions vis a vis the brain and CNS compared to naïve T cells [34].

It is remarkable that BM T cells can respond to bloodborne or CNS-borne antigens in the absence of an adjuvant. This is in contrast to peripheral immune responses, for instance, to cancer vaccines, which require the presence of an adjuvant. Furthermore, the therapy of human tumors in NOD/SCID mice with patient-derived reactivated MTCs from BM was more efficient than with respective MTCs from the patients’ blood [69].

Bloodborne antigens to which T cells can respond include, among others, viral antigens, tumor neoantigens, circulating tumor cells, components released from immunogenic cell death, such as EVs and apoptotic bodies, and immune complexes [34]. Spontaneous induction of cancer-reactive T cells has been deduced from the identification of cancer-reactive MTCs in BM from untreated cancer patients [70]. The repertoire of cancer-reactive MTCs in the BM of such cancer patients was found to be polyvalent and highly individual [71].

Contact of BM T cells with self-antigens from blood or CSF is likely important to maintain self-tolerance and is an example of communication between the BM, blood, and brain to maintain homeostasis.

### 3.5. Communication in the Hippocampus and Choroid Plexus

The CNS and PNS affect the immune system’s impact on immune homeostasis. Likewise, the immune system affects brain homeostasis. Glial cells, as part of the immune system, are specialized for brain-specific functions, including the removal of debris, cleaning (pruning), and the surveillance of synapse function. In addition, the immune system affects the tissue homeostasis of the brain, for instance, via bidirectional communication between the innate immune system and the nervous system for the neuronal homeostasis of the hippocampus [72].

CNS-specific T cells have been reported to shape brain function via the choroid plexus (CP) [73]. The CP is an active neuro-immunological interface that is enriched with CNS-specific CD4+ T cells specific to brain self-antigens. Strategically positioned for receiving signals from the CNS through the CSF and from the immune system through the blood circulation, the CP has an important function in maintaining and restoring brain homeostasis [73].

### 3.6. Neurogenesis in the Hippocampus and Crosstalk with BM-MNCs

The hippocampus is a primary region of neurogenesis within the adult mammalian brain [74]. Its development, physiological regulation, and dysfunction in disease have been described and discussed previously [72]. New are the following findings: (i) Transplantation of bone-marrow mononuclear cells (BM-MNCs) fosters hippocampal neurogenesis and enhancement of neuronal function [75]. Direct transfer of water-soluble molecules from transplanted BM-MNCs to neural stem cells (NSC) in the hippocampus was already observed just 10 min after cell transplantation [75]. (ii) BM-MNCs are able to activate angiogenesis in endothelial cells via gap junction-mediated cell–cell interaction [76]. (iii) An artificial bioactive cell niche can mediate a uniform thermal stimulus for BM-derived mesenchymal stem cells (BM-MSCs). Thermal activation from the matrix to the cells triggered ion channel opening and Ca^2+^ influx, which finally promoted neural differentiation [77]. (iv) Another new platform allowed researchers to study the differentiation of neurons, astrocytes, oligodendrocytes, and microglia from human-induced pluripotent stem cells to form neural tissue-on-chip models [78]. Such models could be used to evaluate the therapeutic potential of EVs derived from BM-MSCs [78]. (v) Recent studies have shown that following neuronal induction, BM-MNCs enter an intermediate cellular state before adopting neural-like morphologies by active neurite extension. This process has been termed BM-MNC neural transdifferentiation [79].

BM-MSCs from calvarial bones could also be involved in hippocampal neurogenesis. Cranial bones constitute a protective shield for the vulnerable brain tissue, bound together as a rigid entity by unique immovable joints known as sutures [80]. Cranial sutures serve as major growth centers for calvarial morphogenesis and have been identified as a niches for MSCs [80].

### 3.7. Neuroimmune–Cutaneous Crosstalk in the Skin and Nose

Anatomical contacts between sensory neurons and skin epidermal cells have been described as an anatomical network for neuroimmune–cutaneous crosstalk [81]. Recent studies have shown that keratinocytes, melanocytes, Langerhans cells, and Merkel cells act as sensory transducers for mechanical, thermal, or chemical stimuli and communicate with intraepidermal free nerve endings via chemical synaptic contacts [81].

Neuronal–immune cell units have been described in allergic inflammation in the nose [82]. Tri-directional neuroimmune networks seem to be involved in allergic inflammation in the nose and in neuroimmune control of the nasal mucociliary immunologically active epithelial barrier [82].

### 3.8. Gut Microbiota–Brain Interactions

Gut microbiota–immune system–brain interactions maintain brain homeostasis and influence brain and behavior [83]. Upon receiving stress signals, the body responds by increasing energy metabolism and initiating immune responses. Corticosterone, a glucocorticoid that regulates secretion along the hypothalamus–pituitary–adrenal (HPA) axis, mediates neurotransmission and humoral regulation [83]. This represents a tripartite regulation between the immune system, the neuronal system, and the hormonal system.

Mounting evidence suggests that communication between the gut and the brain could be key to understanding multiple neuropsychiatric disorders, with the immune system coming to the forefront as an important mediator [84]. Other routes may be involved as well, including the HPA axis, the enteric nervous system, and the vagus nerve [85].

The intestinal tract is the largest immune organ in humans. It comprises a complex network of immune cells and epithelial cells that perform a variety of functions, such as nutrient absorption, digestion, and waste excretion. IL-33 is a newly characterized cytokine which is constitutively expressed in the nuclei of different cell types, such as endothelial, epithelial, and fibroblast-like cells. Upon tissue damage or pathogen encounter, IL-33 is released as an alarmin. It signals through a heterodimer IL-33 receptor (ST2) and has the ability to induce Th2 cytokine production by Th2 cells, mast cells, or basophils. Emerging functions of this pleiotropic cytokine include homeostasis, regulation, and tissue repair [86,87,88].

### 3.9. Summary of Neuroimmune Interfaces

The hippocampus is an ancient, curved structure of the limbic system that helps with learning and memory. New findings demonstrate that BM-MNCs can exert an influence on the hippocampus. For instance, they can foster hippocampal neurogenesis and enhance neuronal function. The direct transfer of molecules from transplanted BM-MNCs to neural stem cells in the hippocampus was already observed just 10 min after cell transplantation. This demonstrates communication between the BM and brain. Another recent study reported on the neural induction of BM-MNC neural transdifferentiation. We are obviously at the beginning of a new neuroimmune communication research field.

Special neuroimmune interfaces facilitate communication between the brain and the immune system. In the brain, a SLYM permits the exchange of molecules between the CSF and venous blood. Also, vascular channels of the inner skull of the cortex allow the passage of cells and CSF-derived antigens and hormones between the skull and the brain parenchyma. Dysfunctions of these brain-draining neuroimmune interfaces might be involved in migraine disorders (see Section 4.5).

The CP is another interface consisting of arteriovenous blood vessel convolutes in the brain ventricles which produce brain liquor. It consists of specialized cells, like epithelium cells, endothelial cells, fibroblasts, and immune cells. The CP is enriched with CD4+ T cells specific to brain self-antigens. It receives signals from the CNS through the CSF and from the immune system through blood circulation. In this strategic position, the CP exerts an important function in brain homeostasis.

Gateway-specific blood vessels represent further neuroimmune interfaces. They occur in the brain and spinal cord, as well as in the retina and in ankle joints. Gateway reflexes have been identified in experimental models of autoimmune encephalitis, lupus erythematodis, autoimmune uveitis, and inflammatory arthritis.

Anatomical contacts have been described between sensory neurons and skin epidermal cells, such as keratinocytes, melanocytes, Langerhans cells, and Merkel cells. They allow neuroimmune–cutaneous crosstalk. Sensory transducers for mechanical, thermal, or chemical stimuli communicate with intraepidermal free nerve endings via chemical synaptic contacts. Neuroimmune networks seem to also be involved in allergic inflammation in the nose.

Via the immune system as a mediator, gut microbiota can influence brain function. This could have positive effects (the maintenance of homeostasis) or negative effects (the development of neuropsychiatric disorders).

The PNS innervates and communicates with bones and the BM. Adrenergic nerve fiber terminals from the sympathicus can exert activating signals, while cholinergic nerve endings from the parasympathicus can exert a dampening effect on stem cells and immune cells in the BM.

A neuro-osteogenic network has been described to control MSC homeostasis. It is of importance for bone regeneration. Without MSCs, there would be no bones and no BM. Nerves from the PNS interact with the bone through innervated axons, multiple neurotrophins, and bone-resident cells.

The main features of Section 3 are shown in Table 2.

### 3.10. A Unifying Hypothesis of CNS-CIS Neuroimmune Homeostasis

A unifying hypothesis from these studies, many of which are from 2023 to 2025, can be constructed with regard to neuroimmune homeostasis. Homeostasis describes a situation of balance of physiological body functions, such as blood pressure, body temperature, blood pH, and other factors. Here, we present a hypothesis of the homeostasis of the brain and immune system. This hypothesis includes everything that is surrounded and protected by bone, namely the CNS and the CIS [34]. Neuroimmune homeostasis depends on the regulation of stem cells that reside in special niches and which are involved in the generation of these tissues. The relevant cells are HSCs for the immune system, NSCs for the CNS, and MSCs for the bones. Homeostatic control of these three stem cells is exerted through a tri-directional neuroimmune network between cells, immune-derived molecules, neuro-derived molecules, and hormones. The hypothesis also states that TNTs and EVs play an important role in neuroimmune homeostasis. TNTs enable intercellular trafficking of organelles, such as mitochondria and lysosomes. They can help neurons in cases of energy crisis or rescue them from aggregate-induced dysfunction. EVs can exchange proteins, membrane components, and cytoplasmic material. Such repair and recycling activity enables efficiency in the use of biological material. EVs can cross the blood–brain barrier, communicate with brain and immune system, and regulate neuroinflammatory processes. Of particular interest are EVs from MSCs, which can dampen neuroinflammation. BM-derived MSCs and MNCs have the potential to foster hippocampal neurogenesis and the enhancement of neuronal function. It is postulated that they and their released TNTs and EVs are part of the neuroimmune homeostasis network. The hypothesis also states that neuroimmune homeostasis functions in an open but autonomous way: the T cell responses from the vertical column as well as those from the skull are independent from classical secondary lymphoid organs. The autonomous system also includes memory B and T cells from specialized niches of the BM which provide long-term survival conditions. In situations of dietary restriction and energy crisis the system functions as a refuge for stem cells and immune memory cells [34,89]. The CNS-CIS neuroimmune homeostasis model includes immune responses to bloodborne as well as CSF-borne antigens.

Figure 1 provides a sketch of the location and function of the brain and immune system. In the real world, their location goes beyond the hippocampus and the BM. NSCs reside in the brain, while HSCs and MSCs reside in BM and solid bone tissue. The hippocampus on top represents an important part of the CNS, while the BM at the bottom represents an important part of the CIS. The main aspect of the figure is a comparison between cognate intercellular communication via the synapses in the brain and the immune system. Synaptic cognate interactions are the basis of cognition, learning, and memory in the brain for the eventual development of consciousness.

## 4. Neuroimmunomodulation

### 4.1. Basic Information

This is a multidisciplinary field which focuses on the way that immune responses are influenced and can be modulated by brain activity and how neuronal function is impacted and can be modulated by immunological signaling [90]. The limbic system (LS) is part of the middle brain (cerebrum) involved in behavioral and emotional responses. It emits endorphins, which are body-made opioids. Melatonin, a neuronal hormone of the pineal gland, exhibits a wide range of physiological functions, such as sleep control, regulation of the circadian rhythm, immune modulation, regulation of mitochondria, autophagy, and metabolism, antioxidant, and anti-aging activity [91]. Melatonin enhances NK cell function in aged mice by increasing transcription factor T-bet expression via the JAK3-STAT5 signaling pathway [92]. Disturbances within the LS can lead to memory dysfunction, depression, anxieties, autism, phobia, narcolepsy, and post-traumatic stress disorder. Neurodegenerative disorders (NDDs) are becoming more prevalent and are now the fourth leading cause of death, following heart disease, cancer, and stroke.

The CNS is the only organ system without lymphatic capillaries. The newly identified glymphatic system provides the CNS with pseudolymphatic activity [93]. Sleep stimulates the elimination actions of the glymphatic system. For instance, the CSF is discharges ß-amyloid as well as tau through the glymphatic system. Glymphatic failure is a potential mechanism in several NDDs, including migraine and age-associated cognitive illnesses. Future research efforts should be targeted to the glymphatic system [94]. Another crucial player in the pathophysiology of NDDs is mitochondrial dysfunction [93]. A comprehensive summary of the key features of NDDs and advances in genome editing in NDDs has been presented as the right path to therapy [95].

The next three paragraphs will deal with non-invasive neurological techniques to treat mental and emotional processes, attentional deficiencies, or major depressive disorders. These promising techniques are mentioned in spite of the fact that molecular mechanisms are not yet available.

### 4.2. Stimulation of the Vagus Nerve

The long vagus nerve is necessary to provide two-way communication between internal body structures and the brain. By promoting interaction with the central, cardiopulmonary, as well as intestinal nervous system, the vagal afferent and efferent nerves of the parasympathetic autonomic nerve system have two distinct roles that impact immunomodulation, enteroendocrine functions, and mental and emotional processes [93]. One way of stimulating the vagus nerve is by electrical signals and is called vagus nerve stimulation (VNS). VNS can dramatically lower the number of activated macrophages and microglia, as well as the level of cytokines associated with inflammation in the brain from mice with lipopolysaccharide-induced inflammation [95].

### 4.3. Neurofeedback

In this modality, electroencephalography (EEG) is the standard technique used to measure cerebral electrical activity. Neurofeedback techniques use attached scalp sensors to collect and capture patterns of brainwave activity. This modality shows promise in improving attentional deficiencies, especially when applied to treat attention deficit hyperactive disorder (ADHD) [95].

### 4.4. Transcranial Magnetic Stimulation (TMS)

Focused magnetic fields, like those in MRI, are utilized in TMS, an innovative, non-invasive neurostimulation technique. Targeting the medial prefrontal cortex (mPFC) in studies on the brain, single-pulse and repeating TMS has demonstrated signs of improving symptoms in major depressive disorders (MDD), post-traumatic stress disorder, and obsessive-compulsive disorder. In addition, TMS has exhibited promise in both diagnosing and managing dementia, particularly in primary degenerative diseases, like Alzheimer’s and vascular dementia [96].

The next three paragraphs deal with the neuroimmunomodulation of selected neuroimmunological disorders. Migraine is an example of MLV dysfunction, multiple sclerosis is an example of a neuroimmune dysfunction, and glioblastoma is an example of neuro-glia dysfunction.

### 4.5. Migraine

Migraine is a disorder characterized by recurring headache attacks accompanied by other neurological symptoms which affects 15% of the world’s population. It ranks second or third worldwide as a cause of disability. In clinical practice, triptans, which are synthetic serotonin receptor agonists, are first-line agents. MLV dysfunction was recently demonstrated to cause migraine-like pain in mice [97]. Trigeminal nerve roots expressing CGRP are in close proximity to CGRP receptor-expressing MLVs. In deep cervical lymph nodes (DCLNs) MadCAM-1-expressing lymphatic endothelial cells interact with CD4+ T cells via integrin alpha4/ß7 to promote pathologic lymphatic vessel remodeling via CGRP signaling. This leads to decreased permeability in MLVs and reduced CSF outflow to DCLNs [97].

Whether such results can be translated to human migraines remains to be seen. Nevertheless, these experimental studies point towards a new mechanism that needs to be elucidated.

### 4.6. Multiple Sclerosis

MS is the most frequent autoimmune disease of the CNS, affecting the brain, spinal cord, and optic nerves. This neurological disease, typically leading to disability among young adults, affects 33 individuals per 100,000 worldwide [98]. Aberrant immune T cells attack a number of myelin and non-myelin antigens leading to demyelinating disease [98]. Self-peptide-dependent autoproliferation of B and T cells is a key mechanism in MS [99]. New data demonstrate an important role of T-bet+CXCR3+ B cells in the pathogenesis of MS both in the peripheral immune system and the CNS compartment [100]. Myelin is necessary for axonal contact formation. Brain-enriched myelin-associated protein 1 (BCAS1) identifies oligodendrocyte lineage cells in the stage of active myelin formation. Such cells are maintained at high densities in the cortex throughout life [101]. Following a demyelinating insult in a mouse model, BCAS1+ oligodendrocytes in remyelinating cortical lesions were recently reported to shift from a quiescent to an activated internode-forming morphology co-expressing myelin-associated glycoprotein MAG [101]. It was suggested that BCAS1+ oligodendrocytes enable efficient cortical remyelination in MS [101]. All approved MS therapeutics work by modulating autoimmune responses.

Cellular stress (e.g., oxidative and nitrogenic stress [102], as well as endoplasmic reticulum stress [103]) plays an important role in the pathogenesis of autoimmune models of MS. Altered redox homeostasis and increased oxidative stress are a trigger for the activation of a brain stress response [102]. This can be modulated by theta burst stimulation [102] and by vitagenes [103]. Mechanisms controlled by vitagenes operate in the brain for the maintenance of redox homeostasis. Vitagenes encode for heat-shock proteins (HSP), namely HSP32 and HSP70, and for thioredoxin the sirtuin protein systems [104].

Experimental autoimmune encephalomyelitis (EAE) mouse models have been useful to elucidate molecular mechanisms of MS and to develop potential immunomodulatory drugs. Based on such studies, novel therapeutic targets have been suggested, such as the Th17 response [105,106] and stress granules [107]. Antioxidative stress molecules, such as panobinostat [108], MitoQ [109], siponimod [110], TFM-735 [111], melatonin [112], vitamin D [113], alpha-lipoic acid [114], atrine [115], and fingolimod [116] showed attenuating effects in murine EAE models.

In spite of such experimental studies there still exists no standard therapy for MS. One question relates to the relevance of the murine EAE model for human MS. As a neuroautoimmune disease, MS resembles a complex puzzle in which cellular and humoral immune responses are involved. The humoral immune response involves B cells, antibodies, and complement components. New results from 127 patients with MS demonstrate that aberrant complement activation (in particular C4a) is strongly associated with structural brain damage revealed by quantitative MRI metrics [117]. NfL as a marker of neuroaxonal damage was strongly associated with C4a, C3a, Ba, and Bb. GFAP as a marker of astrocyte damage was associated with C1q and with progression in MS and accelerated gray matter loss [117]. Other studies revealed a lymphocyte–microglia–astrocyte axis [118] and a C1q-C3 complement axis associated synaptic changes in the hippocampal neurons of patients with MS [119].

### 4.7. Glioblastoma: Neuroimmunomodulation and Immunotherapy

GBM is the most frequent primary malignant brain tumor in adults, with the worst prognosis. The incidence is 4 to 5 patients per 100,000 adults per year [120]. Glioma stem-like cells (GSCs), vascularization, hypoxia, metabolic reprogramming, tumor-promoting inflammation, and sustained proliferative signaling are hallmarks of GBM development [121]. The following paragraphs provide examples of neuroimmunomodulation (i)–(iv) and immunotherapy (v).

(i)Inhibition of neuron–glioma interaction. With regard to intercellular communication there is one key feature which is unique to GBM in comparison to other cancers, namely neuron–glioma interaction. Interactions between presynaptic neurons and postsynaptic gliomas drive GBM tumor development. The interactions involve paracrine signaling factors (e.g., neuroligin-3, brain-derived neurotropic factor 1–3), GABAergic synaptic communication, and α-amino-3-hydroxy-5-methyl-4-isoxalol-propionic acid (AMPA) postsynaptic currents [121]. GABAergic synaptic communication can be inhibited by levetiracatam and AMPA postsynaptic currents can be inhibited by by perampanel [121].(ii)Antidepressants. Neuroimmunomodulation and the treatment of GBM is complex and influenced in a dynamic way by tumor-host interactions, in particular in the tumor microenvironment (TME). For instance, GSCs home in on a specific TME niche consisting of stromal and immune cells with many reciprocal intercellular communications. GSCs communicate with their TME by cell–cell interaction via TNTs [121]. Antidepressants, such as imipramine, amitryptyline, fluoxetine, mirtazapine. agomelatine, and escitalopram, are prescribed to inhibit GSC plasticity and to combat the side effects of chemotherapy [121].(iii)Anti-hypoxia treatment. Hypoxia leads to autophagy which inhibits the effects of radio- and chemotherapy. Chloroquine treatment can inhibit autophagy. The monoclonal antibody bevacizumab can inhibit the angiogenesis factor vascular endothelial growth factor (VEGF). Hypoxia also upregulates the hypoxia-inducible factor HIF-1α which drives cellular metabolism toward anaerobic fermentation. Mebendazole and melatonin can be used to normalize HIF-1α expression levels [121]. Interestingly, oncolytic Newcastle disease virus (NDV) was demonstrated to be capable of breaking hypoxia and other cancer resistances [122].(iv)Anti-immunosuppression. Due to metabolic reprogramming and TME acidification, M2 macrophages and glia-associated macrophages (GAMs) are upregulated, which inhibits CTL responses. The upregulation of TGF-ß by both tumor-associated macrophages (TAMs) and tricarbonic acid (TCA) cycle mutations further inhibit the innate and adaptive immune system. TCA cycle mutations can be inhibited with ONC201 treatment, isoelectric inhibitors, and peptide vaccines. Anti-PDL1 checkpoint inhibitors can inhibit immunosuppression exerted through PDL1+ M2 macrophages [121].(v)Active-specific immune stimulation/immunotherapy (ASI). The Immune-Oncological Center Cologne, Germany (IOZK), has developed an individualized multimodal immunotherapy (IMI) for cancer patients in which oncolytic NDV and cancer-derived EVs play an important role. The scientific rationale and clinical experience has recently been summarized [50]. The strategy involves repeated cancer-immunity cycles evoked in cancer patients by systemic NDV exposure combined with modulated electrohyperthermia (mEHT) pretreatment to induce immunogenic cell death (ICD). This “ICD immunotherapy” generates cancer cell debris, including EVs and apoptotic cell bodies that accumulate in blood plasma. These immunogenic components, carrying important information about the individuality of the patient’s cancer (e.g., neoantigens and shared TAs), are harvested and loaded onto patient-derived DCs to generate the dendritic cell vaccine IO-VAC^R^. This is then used for ASI by intradermal vaccination. The IMI strategy involves three treatment phases: (I) Anticancer treatment (resection, radio- and chemotherapy, and ICD immunotherapy), (II) immunization (ICD immunotherapy, ASI, and neuroimmunomodulation), and (III) maintenance and immune protection (ICD immunotherapy, peptide vaccines, boosted DC vaccines). Such a combined treatment strategy should be the aim for future GBM treatment approaches [121].

A recent retrospective analysis from a group of n = 71 adults with isocitrate dehydrogenase 1 (IDH1) wild-type GBM treated with standard therapy plus IMI revealed a 2-year overall survival (2y OS) of 42.7% for MGMT-promoter unmethylated types and 75.5% for MGMT-promoter methylated types [50]. In comparison to standard therapy without IMI, the data demonstrate an increase in 2y OS by a factor of 1.5 (unmethylated) and 2.9 (methylated). The standard of care of GBM is strengthened and improved with IMI; thus, this treatment principle is continued at IOZK [50].

The positive findings of the IMI strategy are corroborated by results from a phase 3 prospective externally controlled cohort trial. Association was reported between autologous tumor lysate-loaded dendritic cell vaccination and the extension of survival among patients with newly diagnosed and recurrent GBM [123].

### 4.8. Summary

In summary, a variety of neuroimmune disorders and psychiatric illnesses are amenable to neurological or immunological manipulation. Electrical vagus nerve stimulation can lower inflammation in the brain. Transcranial magnetic stimulation can improve MDDs. Immunomodulatory drugs offer prospects for the management of autoimmune neurological diseases, such as MS. Sophisticated forms of active–specific immunotherapy and other means are a new modality to treat brain tumors, such as GBM. The three examples of dysregulation of neuroimmune homeostasis represent different levels of development of neuroimmunomodulation. Studies on migraines are just beginning. Studies on MS are further advanced, but no standard protocol of treatment has been reached. Studies on GBM have reached the evidence level of a phase 3 prospective trial. They have achieved the transition from immunomodulation to immunotherapy. The main features of Section 4 are shown in Table 3.

## 5. Conclusions and Perspectives

The brain has been thought for decades to be an immune-privileged organ, meaning that the immune system has no access to this organ because of the blood–brain barrier. Neuroimmunology and psycho-neuroimmunology were, for a long time, not considered as basic scientific disciplines. However, this view has changed. Brain borders are at the center stage of neuroimmunology. Ongoing research and the development of new technologies allow progressively deeper insights into the nervous system and into the immune system, into the interactions between the two systems, and into new concepts of neuroimmunomodulation.

The following are a few examples of the progress in neurosciences since the year 2000: (i) signal transduction in neuronal synapses through dopamine (Nobel Prize in Medicine 2000 to A. Carlsson and P. Greengard); (ii) magnetic resonance imaging (MRI) (Nobel Prize in Medicine 2003 to P. Mansfield and P. Lauterbur); (iii) the discovery of olfactory receptors (Nobel Prize in Medicine 2004 to R. Axel and L. Buck); (iv) the first profound stimulation of the brain to treat obsessive compulsive disorders (2008); (v) optogenetics allowing the stimulation by light of certain predetermined neurons of the brain (2010); (vi) the discovery of spatial position neurons in brain (Nobel Prize in Medicine 2014 to M.-B. Moser and E. Moser); and (vii) the discovery of receptors for temperature and touch (Nobel Prize in Medicine 2021 to D. Julius and A. Patapoutian).

Examples of progress in immunology since 2000 also include the following: (i) discoveries concerning the activation of innate immunity (Nobel Prize in Medicine 2011 to B.A. Beutler and J.A. Hoffmann) and the discovery of the dendritic cell and its role in adaptive immunity (to R.M. Steinman); and (ii) the discovery of cancer therapy by the inhibition of negative immune regulation (Nobel Prize in Medicine 2018 to J.P. Allison and T. Honjo).

The following important discoveries in general cell biology are also of relevance to this review: (i) The discovery of a machinery regulating vesicle trafficking, a major transport system in cells (Nobel Prize in Medicine 2013 to J.E. Rothman, R.W. Schekman and T.C. Südhof), (ii) the development of a nanoscope to study the function of cellular nanostructures (Nobel Prize 2014 to E. Betzig, W.E. Moerner and S.W. Hell), and (iii) the discovery of microRNA (Nobel prize in Medicine 2024 to V. Ambros and G. Ruvkim).

In addition, it is worth mentioning a few neuroscientific and immunological tools that were used for new discoveries, such as the following: (i) nuclei isolation from brain tissue samples and generation of single-nucleus multiome data, (ii) transcription factor motif enrichment analysis, (iii) neighborhood enrichment and intercellular communication modeling, (iv) quantification of ligand–receptor communication using CellChat, (v) gene regulatory network analysis, (vi) single-cell RNA-seq analysis of glial progenitor differentiation, (vii) hippocampal subregion dissection and synaptosome preparation, (viii) proteomics, (ix) the use of congenic mouse strains, (x) the introduction of new labels into T cells through genetic manipulation, (xi) TCR sequencing, (xii) the Polylox system to introduce unique labels into T cells, (xiii) the adoptive transfer of single T cells and flow cytometric analysis, and (xiv) the use of gene knockout mice for the functional analysis of single genes. Such technologies are fundamental for better understanding of the principles of homeostasis in the brain and immune system. This then allows researchers to target pathologies that can arise from the dysfunction of intercellular communication.

The brain and immune system show similarities in their developmental and organizational principles. Tri-IPC precursor cells lead to the development of neurons, oligodendrocyte precursors, and astrocytes. Common lymphoid precursor cells (CLPs) lead to the development of lymphocytes, myeloid precursors, and mast cells. The local environment is constantly surveyed by astrocytes (brain) or DCs (body tissues) which then crosstalk with neurons (brain) or T lymphocytes (body tissues). Synapses in the brain and immune system include tripartite cellular interactions. The homeostasis of neurons involves excitatory and inhibitory protein communities, while the homeostasis of T cells involves signals from positive and negative signaling receptors.

The principles of intercellular communication in the brain involve electrical signals, primarily via neuronal synapses, where they are translated into chemical signals (e.g., neurotransmitters and gliotransmitters). Cognate intercellular interactions in the immune system involve immune synapses. These directly transmit chemical signals and do not involve electrical signals. Further communication in the immune system occurs via a multitude of cell surface receptors interacting with corresponding ligands, such as cytokines and interferons. An important task of the immune system is to distinguish between self and foreign antigens and between signals mediating tolerance or danger. Both the innate and adaptive immunity systems contribute to decision making within the immune system. A specialty of the adaptive immunity system is the process of somatic gene rearrangement. This allows each B or T lymphocyte to express a unique antigen-specific receptor at its cell surface. These receptors are pre-selected in lymphatic organs (thymus, BM) against self-reactivity. The repertoire of B and T cell receptors and of immune synapses is enormous. Any foreign chemical structure, be it natural or artificial, can become recognized to elicit an immune response. The repertoire of neuronal synapses is, likewise, enormous.

Neuroimmune networks and interfaces allow communication between the brain and the immune system. The nervous system regulates homeostasis within the immune system through neurotransmitters and neuropeptides. Conversely, immune cells survey the brain for homeostasis, for instance via gliotransmitters. Glia represents the immune component adapted to the requirements of brain. Gliotransmitters from astrocytes are involved in synapse communication. Microglia are involved in pruning and can rescue neurons from aggregate-induced neuronal dysfunction and death. CNS-specific CD4+ T cells specific to brain self-antigens shape brain function via the CP. Special neuroimmune interfaces facilitate communication between the brain and immune system, e.g., SLYM, CP, and gateway reflexes. Neuroimmune–cutaneous crosstalk and a neuro-osteogenic network also exist. Bidirectional communication also occurs between the brain and BM, while the gut microbiota can influence brain function via the immune system as a mediator. Homeostasis at these interfaces is important to prevent dysfunctions, which can lead to severe neuropsychiatric illnesses.

Intercellular communication in the brain and immune system also involves TNTs and EVs, structures which have only recently been discovered. TNTs allow cells to connect at short or long distances and to exchange, astonishingly, whole organelles, such as lysosomes and mitochondria. Neurons and immune cells also release EVs. These may play an important role in intercellular and interorgan communication as well as in cellular repair and the recycling of cytosolic proteins and membrane components. EVs also interact with the complement system in neurological diseases. Finally, EVs can be used for cancer immunomodulation and cancer immunotherapy, as exemplified by the IMI strategy of IOZK, Cologne, Germany.

Experimental approaches and technologies towards TNTs and EVs are at their initial stages and require standardization and evaluations before conclusions can be drawn about their real impact and relevance. Nevertheless, the reports that these extracellular structures can exchange cellular material between the brain and immune system suggest that nature has invented until-now unknown mechanisms to increase efficiency in the use of biologic material.

The new hypothesis about CNS-CIS neuroimmune homeostasis is focused on everything that is associated with and protected by bones, including the brain with the CNS, BM from the skull to the pelvis (Figure 1) as the CIS, and solid bones as supporting structures. It includes self and non-self-antigens from the blood and CSF for immunosurveillance. In addition, the hypothesis suggests that TNTs and EVs have the potential to communicate, to repair, and to help in situations of neuroimmune stress and crisis. The unifying hypothesis states that homeostasis of the brain and immune system depends on the regulation of the neuronal and hematopoietic stem cells that are involved in the generation of these tissues. They also depend on the regulation of MSCs, which generate the solid bone structure of vertebrates. Calvarial bone and its BM protects the brain. The bone of the vertebral column with its BM protects the lower part of the CIS (see Figure 1). Specialized niches in BM provide optimal conditions for long-term survival not only for stem cells but also for immune memory cells. Memories within the neuroimmune system are autobiographical (brain), polyvalent, and highly individual (BM). The hypothesis also states that neuroimmune homeostasis functions in an open but autonomous way. Relay stations connect the central systems with the peripheral ones. CIS and PNS function in an autonomous or semi-autonomous way. Regulatory mediators are neuro-derived molecules (neurotransmitters or neuropeptides), immune-derived molecules (cytokines, chemokines), and hormones produced by endocrine glands. The hypothesis emphasizes the importance of homeostatic control between NSCs, HSCs, and MSCs.

That the neuroimmune homeostasis might function in an autonomous way is highly speculative. However, it might explain how the system could have survived hundreds of millions of years under constantly changing environmental conditions. Research on the three stem cell types and on TNTs and EVs provides promise and a good perspective for the future. Research on BM immune responses might, in future, explain why these responses do not depend on adjuvants as the corresponding responses from the periphery. The superiority of cancer-reactive MTCs from BM in comparison to blood, proven in 2001 [68a], has, unfortunately, not yet resulted in corresponding research and clinical activities.

Studies on BM are technically difficult because of the solid bone structure that protects this tissue. Nevertheless, because of conviction, the author and colleagues have studied BM immune responses for 31 years in animal models and cancer patients and published novel findings in international peer-reviewed journals, including *Nature Medicine*, totaling 19 papers. A critical comment, therefore, may be allowed concerning the presentation of BM in textbooks of immunology. In spite of its importance as a central organ with the capacity of primary and secondary immune functions, as known for more than 20 years, textbooks still present the old-fashioned view of BM as being just a primary immune organ.

Any component in the complex cellular communication and interaction system between the brain and immune system can lead to dysfunction. Three examples have been selected, namely migraine, MS, and brain cancer. In migraine, which affects 15% of the world’s population, MLV dysfunction was shown in a mouse model to lead to decreased permeability and reduced outflow of CSF to deep cervical lymph nodes. MS is an example of a deregulated immune response. As an autoimmune neurological disease, MS affects the brain, spinal cord, and optic nerves. Three main components lead to MS, namely (i) cellular stress in oligodendrocytes, (ii) Th17 immune T cells and B cells attacking myelin-associated proteins, and (iii) an aberrant complement reaction possibly mediated via EVs. Neuroimmune modulatory strategies in MS involve immunosuppressive drugs, antioxidant reagents (e.g., melatonin, vitamin D, MitoQ, and panobinostat) and complement inhibition as a potential future therapeutic option. A third example of neuroimmune dysfunction is malignant brain tumor, such as GBM, with the worst prognosis. Due to its complexity and dynamic tumor–host interactions, in particular in the TME, standard therapy by surgery and radiochemotherapy is insufficient. Thus, there is a need for innovative treatments, for instance multiphase individualized combination treatments. One such strategy, IMI, is explained, and the positive findings obtained at a private institution have been mentioned. In spite of the fact that evidence has been obtained for the effectiveness of post-operative vaccination from a randomized controlled GBM clinical study, such results need to be reproduced and extended in further clinical studies.

Significant advancements have been made in recent decades in the fields of neuroscience and immunology. These now enable selective targeting of dysfunctional molecules which need to be inhibited (e.g., in MS), activated (e.g., in GBM), or just modulated (e.g., neuroimmunomodulation).

In terms of evolution, the brain and immune systems went through an ancient archaic phase in which multicellular organisms appeared on earth. The brainstem and the innate immune system were invented by nature and selected in ancient times. Later, about 500 million years ago, vertebrates developed increasingly sophisticated brain structures and also the adaptive immunity system. That both the brain and immune system survived for such a long time period under constantly changing environmental conditions must be declared as a success story.

## Figures and Tables

**Figure 1 ijms-26-06552-f001:**
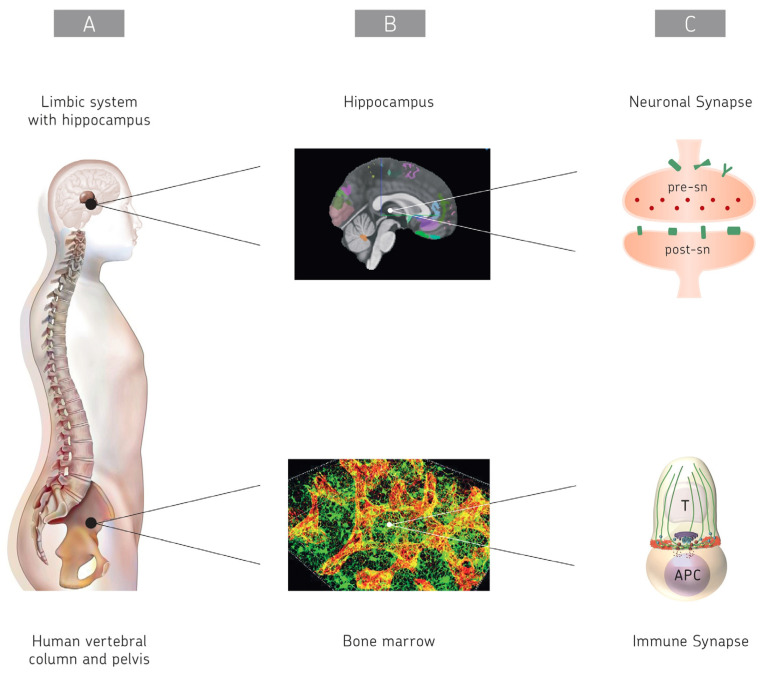
(**A**) A representation of the human body with the head, including the brain with hippocampus as the central nervous system and the vertebral column and pelvis including BM as the central immune system. The iliac crest of the pelvis is often used to extract samples of BM for clinical studies. (**B**) shows the top of the hippocampus region of the brain, which is important for learning and memory and contains neuronal stem cells. The bottom part shows the spongy architecture of the BM. Sinusoidal vessels are stained in red and perivascular sinusoidal parenchyma are stained in green. The green areas contain niches for HSCs, MSCs, and for memory B and T lymphocytes (image: https://ashpublications.org/ (accessed on 8 July 2023)). (**C**) provides a top-down sketch of a neuronal synapse with a presynaptic neuron (pre-sn) containing vesicles with dopamine, acetylcholine, glutamate, GABA, or neuropeptides and a postsynaptic neuron (post-sn) expressing cell surface receptors specific to the synapse-released neurotransmitter or neuropeptides. The power of the neuronal impulse, expressed by the frequency of the action potential, is transmitted chemically via the concentration of the neurotransmitter. This chemical transformation process is associated with the opening of calcium channels in pre-sn and of sodium channels in the post-sn. The bottom part provides a sketch of an immunological synapse with a polarized T cell upon cognate interaction with an antigen-presenting cell (T-APC). Cytoskeletal microfilaments (green), centrosome polarization and signaling events transmitted via signaling complexes occur at the interaction platform (red) (image: https://www.frontiersin.org/ (accessed on 8 July 2023)).

**Table 1 ijms-26-06552-t001:** Presents some typical features discussed in Section 2.

Feature	Description	Ref	Year
Synapses in the brain	One neuron can make up to ten thousand synapses.	[6]	2023
	The brain comprises more than 1 × 10^14^ synapses.		
Neurogenesis	Tri-IPC cells lead to neuronal development.	[19]	2024
Immune synapses	The repertoire of α and β T cell receptors is about 10^16^, while that of γ and δ T cells is about 10^18^. The B cell receptor repertoire is about 10^11^.	[4]	2021
Repertoire of antigen-specific immune receptors	These numbers give an impression of the diversity of immune synapses.		
Lymphopoiesis	Lymphocyte receptor rearrangement occurs in CLP cells.	[33]	2010
Neurotransmitters	Serotonin, epinephrine, dopamine, acetylcholine, and gamma-aminobutyric acid	[1]	2021
		[10]	1996
Gliotransmitters	Glutamate, d-serine, ATP	[2]	2016
Cytokines	Il-1 to Il-33	[4]	2021
Interferons	IFN-α, β, and γ	[4]	2021
Synaptic ectosomes	Exocytosis via Sorting nexin 27	[43]	2022
		[52]	2024
Gap junctions	Transmission of small molecules, calcium flux	[1]	2021
TNT	F-actin containing thin nanoprotrusions	[22]	2021
EV	Extracellular vesicle	[28]	2023
	Interorgan crosstalk	[27]	2019

CLP = Common lymphoid progenitor; Il-1 to Il-33 = interleukins; IFN = interferon, TNT = tunneling nanotube; EV = extracellular vesicle.

**Table 2 ijms-26-06552-t002:** Neuroimmune interfaces and network communications.

Feature	Description	Ref	Year
SLYM	Permits in-brain exchange of solutes between CSF and venous blood; lymphatic-like membranes and vascular channels; glymphatic system	[58]	2023
		[60]	2025
Gateway reflexes	Specific vessels at distinct sites	[61]	2023
Brain and bone marrow	Adrenergic and cholinergic nerves running in BM adjacent to arteries and arterioles	[65]	2023
		[34]	2023
	A neuro-osteogenic network	[66]	2023
	Bone regeneration	[67]	2023
Choroid plexus	An active neuro-immunological interface	[73]	2013
Hippocampus	Bidirectional neuroimmune communication for homeostatic neurogenesis	[74]	2015
	Crosstalk with BM-MNCs	[75]	2024
Intraepidermal free nerve endings in skin	Neuroimmune–cutaneous crosstalk	[81]	2023
	Network in nose allergic inflammation	[82]	2022
Gut–brain interactions	Gut microbiota–immune system–brain interactions	[83]	2023
	Depressive disorders	[84]	2024
	Role of IL-33	[86]	2023
CNS-CIS neuroimmune homeostasis hypothesis	Regulatory control over three types of stem cells, namely HSCs, NSCs and MSCs	This review	2025
	The supporting role of TNTs and EVs		
	CIS: Storage and refuge for immune memory	[89]	2019
		[34]	2023

SLYM = Subarachnoid lymphatic-like membrane; CSF = cerebrospinal fluid; BM = bone marrow; BM-MNCs = bone marrow mononuclear cells; HSC = hematopoietic stem cell; NSC = neuronal stem cell; MSC = mesenchymal stem cell.

**Table 3 ijms-26-06552-t003:** Neuroimmunomodulation.

Feature	Description	Ref	Year
Vagus nerve stimulation	Modulation of enteroendocrine functions and mental and emotional processes	[95]	2023
Transcranial magnetic stimulation	Modulation of major depressive disorders, dementia, or degenerative diseases	[96]	2014
Active–specific immune stimulation (ASI)	Cancer: GBM immunotherapy (IMI)	[50]	2024
GBM	ICD immunotherapy	[121]	2025
	Inhibition of neuron-glioma interaction	[121]	2025
		[121]	2025
	Antidepressants	[121]	2025
	Anti-immunosuppression		
	Anti-hypoxia treatment	[121]	2025
	Breaking therapy resistance	[122]	2019
	Phase 3 clinical study	[123]	2023
Multiple sclerosis:	Th17 cells directly harming oligodendrocytes	[105]	2021
-Aberrant T cell response			
-Immunomodulation	T cell targets: a number of myelin and non-myelin antigens. T cell–B cell interactions	[100]	2025
	BCAS1+ oligodendrocytes for remyelination	[101]	2024
	TIA1-mediated stress granules	[107]	2025
	Redox regulation of cellular stress in MS	[103]	2011
-Aberrant complement activation	Association with structural brain damage	[117]	2025
Migraine	MLV dysfunction	[97]	2024
	Triptans		

ICD = immunogenic cell death; IMI = individualized multimodal immunotherapy; GBM = glioblastoma multiforme; BCAS1 = brain-enriched myelin-associated protein 1.

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
