# Peer review of "Brain and Immune System: Intercellular Communication During Homeostasis and Neuroimmunomodulation upon Dysfunction"

_ijms, 2025, doi:10.3390/ijms26146552_

Round 1
Reviewer 1 Report
Comments and Suggestions for Authors
The manuscript presents a detailed and well-researched review of how cells in the brain and immune system communicate, discussing key mechanisms such as synaptic transmission, gap junctions, TNTs, EVs, and immunological synapses. It also explores neuroimmune interfaces like SLYM, gateway reflexes, and brain–bone marrow interactions, providing insight into how disruptions in these pathways contribute to neurological disorders such as migraine, multiple sclerosis, and glioblastoma. By addressing the complex crosstalk between the nervous and immune systems, the review highlights an increasingly important research area with significant implications for neurodegenerative and neuroinflammatory diseases.
However, the review would benefit from a clearer statement of its unique contribution and criteria for selection, streamlined organization to reduce excessive descriptive detail, and enhanced critical analysis that integrates the discussed mechanisms into a unifying framework. Additionally, inclusion of figures, improved tables, consistent terminology, and refined formatting are needed to strengthen the manuscript's overall impact. Overall, the review has significant potential and covers a highly relevant topic.
Here are my detailed comments and suggestions:
- The review discusses multiple disorders (migraine, multiple sclerosis, glioblastoma, etc.), but the criteria for selecting these particular conditions are not justified. Explaining why these examples were selected would best illustrate the review’s focus and strengthen the
- Certain sections (e.g., Section 2.1 on neuronal intercellular signal communication and the extensive description of immune synapse formation) are very detailed. Consider streamlining some parts by focusing on the most relevant mechanisms rather than exhaustive textbook-level details.
- The transitions between sections, especially when moving from brain-specific communication to immune processes seems little abrupt. A brief integrative summary at the end of major sections that highlights the common principles or differences may improve the flow.
- Some headings are too descriptive (e.g., “2.2.11. Intercellular communication in the immune system by TNTs and EVs”) and could be condensed for clarity. Consider grouping similar subsections under broader headings with shorter titles.
- Although many mechanisms are described, the review sometimes stops at description rather than offering critical analysis. For example, in discussing TNTs and EVs, it would be useful to evaluate the strengths and limitations of current experimental approaches and highlight controversies in the field.
- The review would benefit from a dedicated section that synthesizes the information into a unifying model or framework. This could help clarify how intercellular communication in both systems can be targeted for neuroimmunomodulation.
- While the review covers both molecular mechanisms and clinical conditions, the connection between the two is not always clear. Please try to strengthen the discussion on how these molecular insights translate into therapeutic strategies.
- The manuscript does not describe the criteria for literature inclusion (e.g., search strategy, databases used, inclusion/exclusion criteria). Please include a brief section or at least a paragraph on “Methods” explaining how the literature was selected.
- Some complex concepts and pathways would be better illustrated through well-designed diagrams. Consider adding figures that integrate major mechanisms and interactions discussed in the review.
- There are inconsistencies in the use of abbreviations (e.g., “TNT” vs. “tunneling nanotubes,” “EV” vs. “extracellular vesicles”). Please ensure uniform terminology throughout the manuscript
- A careful proofreading is needed to correct minor typos, inconsistent spacing, and punctuation errors, particularly in the reference citations.
- Some sections rely heavily on a few key references. Consider incorporating a broader range of sources to present multiple perspectives and a balanced discussion.
Author Response
I thank the reviewers for their constructive comments. In the revised version, sentences that should
be removed are highlighted in blue. New sentences are highlighted in red.
Reviewer 1.
1. The criteria for selecting those disorders have been specified in the second paragraph of the
introduction. I) Relevance: Worldwide prevalence 15% (migraine), most frequent autoimmune disease of the CNS (MS), most frequent primary brain tumor (GBM). Ii) Examples for MLV dysfunction (migraine), neuro-immune dysfunction (MS), neuro-glia dysfunction (GBM).
2. Streamlining has been done in section 2.1.4. and in 2.2.7. and in 2.2.11.
Text to be removed is highlighted in blue.
3. Brief summaries have been included in 2.1., 4.1., 4.4., and 4.7.
4. Headings have been shortened in sections 2.1.5., 2.1.6., 2.2.2., 2.2.4., 2.2.8.,
2.2.9., and 2.2.11.
5. A paragraph has been added to this point at page 23.
6. A unifying hypothesis of homeostasis is presented in section 3.9 and in Fig. 1
7. One paragraph is included in section 4.1. stating that the three mentioned physical treatments
(4.2. until 4.4.) are useful even though molecular mechanisms are still lacking. In all other
neuroimmunomodulatory approaches molecular mechanisms have been mentioned
wherever possible.
8. The criteria for literature selection have been mentioned in the last paragraph of the
Introduction.
9. Figure 1 integrates major mechanisms and interactions discussed in the review. It shows
brain with hippocampus and immune system from bone marrow and presents cartoons
of neuronal and immunological synapses. These two locations (hippocampus and bone
marrow) of synapses are just exemplary. There are plenty more locations in brain and
immune system.
10. Inconsistencies of abbreviations have been removed.
11. A careful proofreading has been done
12. 8 new key references have been added to broaden the range of sources.
I believe that thanks to the comments of the reviewers the manuscript has
Improved considerably.
Reviewer 2 Report
Comments and Suggestions for Authors
Dear Authors
There are so many literatures about "Brain and immune system"
1. What's the difference in your manuscript compared with previous reviews about "brain and immune system"?
2. What's the reference of this sentence? The review compares principles of organization of brain and immune system, two important organs developed during 500 million years in multicellular organisms -- This sentence in the abstract has no meaning.
3. Please, rephrase this sentence "Deep insights into such neurological and/or immunological dysfunctions and technological advances in neurosciences and immunology will be shown to enable neuroimmunomodulation and the development of new treatments targeted to distinct molecules"
4. Please, draw at least two figures to help the readers understand.
5. The contents in all Tables were so broad and primitive, and they had no detailed explanation. These kinds of tables had no meaning. The authors should provide detailed information in all Tables.
Author Response
I thank the reviewers for their constructive comments. In the revised version, sentences that should
be removed are highlighted in blue. New sentences are highlighted in red.
1.This question has been answered in the new last paragraph of the Introduction.
2.Many abstracts have an introductory sentence to the topic.
3.The sentence has been rephrased.
4.One new figure has been conceived that will help readers to better understand.
5.Table 4 at the end has been removed. The other three tables provide detailed
information via the cited references. Also, new details have been added.
I believe that thanks to the comments of the reviewers the manuscript has
Improved considerably.
Reviewer 3 Report
Comments and Suggestions for Authors
The review article by Volker Schirrmacher, titled "Brain and Immune System: Intercellular Communication During Homeostasis and Neuroimmunomodulation Upon Dysfunction", compares the organizational principles of the brain and immune system. It highlights that many intercellular communication mechanisms, such as synapse formation, exhibit significant similarities. Both systems are deeply interconnected to protect the body from external and internal threats, maintaining regulatory balance during homeostasis. Three examples (migraine, multiple sclerosis, and brain cancer) are selected to illustrate homeostasis dysfunction.The manuscript is ambitious through the lens of intercellular communication. While the title is attention-grabbing, the article’s specific contribution to the current scientific niche in this field remains unclear. To the present, exist many reviews on neuroimmune crosstalk.
The article becomes tedious due to overly descriptive generalizations that reiterate well-established knowledge, which is not ideal for a review. A review should focus on novel aspects, such as newly discovered cellular communication pathways, rather than redundantly rehashing superficial or widely known information. Early sections (brain, immune synapses) are extremely detailed, whereas later disease‑focused sections (GBM immunotherapy, MS complement) feel more like lists of examples without critical synthesis of conflicting results or clinical trial outcomes.
The author is advised to reorganize the article to emphasize novel molecular insights, eliminate repetitive sections, and avoid presenting general information as if it were a textbook chapter. Under these conditions, I do not consider the article ready for publication.
Author Response
The review article by Volker Schirrmacher, titled "Brain and Immune System: Intercellular Communication During Homeostasis and Neuroimmunomodulation Upon Dysfunction", compares the organizational principles of the brain and immune system. It highlights that many intercellular communication mechanisms, such as synapse formation, exhibit significant similarities. Both systems are deeply interconnected to protect the body from external and internal threats, maintaining regulatory balance during homeostasis. Three examples (migraine, multiple sclerosis, and brain cancer) are selected to illustrate homeostasis dysfunction.The manuscript is ambitious through the lens of intercellular communication. While the title is attention-grabbing, the article’s specific contribution to the current scientific niche in this field remains unclear. To the present, exist many reviews on neuroimmune crosstalk.
The article becomes tedious due to overly descriptive generalizations that reiterate well-established knowledge, which is not ideal for a review. A review should focus on novel aspects, such as newly discovered cellular communication pathways, rather than redundantly rehashing superficial or widely known information. Early sections (brain, immune synapses) are extremely detailed, whereas later disease‑focused sections (GBM immunotherapy, MS complement) feel more like lists of examples without critical synthesis of conflicting results or clinical trial outcomes.
The author is advised to reorganize the article to emphasize novel molecular insights, eliminate repetitive sections, and avoid presenting general information as if it were a textbook chapter. Under these conditions, I do not consider the article ready for publication.
Response to Reviewer 3
I thank the reviewer for his/her time and effort to evaluate the manuscript. His/her comments have been a challenge. They were helpful to improve the manuscript and its specific profile. Please find in the revised version sentences highlighted in blue which should be removed. Sentences highlighted in red are new and should be included.
Point 1: The articles specific contribution. With regard to neuroimmune homeostasis a new hypothesis is being presented. It is based on the involvement of three types of stem cells (HSCs, NSCs and MSCs). Control is exerted through the already known tridirectional neuroimmune network. New is the inclusion of TNTs and EVs. They allow intercellular trafficking of organelles such as mitochondria and lysosomes. EVs can exchange proteins, membrane components and cytoplasmic material and thus contribute to repair and recycling mechanisms. EVs from MSCs can dampen neuroinflammation.
Bone marrow derived MSCs and mononuclear cells (BM-MNCs) have the potential to foster hippocampal neurogenesis and enhancement of neuronal function. In this model they are part of the neuroimmune homeostasis network.
Figure 1 is also novel. It compares structure and function of hippocampus and of bone marrow with histological and molecular insights and detail.
Point 2: Textbook knowledge has been reduced.
Point 3: Novel aspects. The revised manuscript contains 119 references of which 38% are from 2023 to 2025. Perhaps this time window can be considered as new or novel. Purpose of a review. A review is not an original manuscript. Novelty is one thing but a review such as this is another thing. This review is directed to readers in two different scientific disciplines. It is therefore important to point out historically important developmental steps in neuroscience and immunology.
Point 4: The presentation and interpretation of the three sections on neuroimmunomodulation and immunotherapy has been reorganized. The three examples of dysregulation of neuroimmune homeostasis represent different levels of development of neuroimmunomodulation. Studies on GBM have reached an evidence level of a phase 3 prospective trial.
Point 5: Repetitive sections have been eliminated and novel molecular insights (e.g. TNTs and EVs) have been emphasized.
If there still exist doubts about the novelty of this review please let me know by citing one or more respective references for prove.
I thank again the reviewer of his/her efforts and contribution. The comments helped considerably to improve the manuscript.
Sincerely,
Volker Schirrmacher

Round 2
Reviewer 2 Report
Comments and Suggestions for Authors
Dear Authors
The author did not follow MDPI's policy in the revised manuscript.
The author should respond to each of the reviewer’s comments individually, linking their responses clearly to the corresponding questions.
1. The authors did not respond well to the reviewer's feedback.
"The profile of the review differs from other publications about brain and immune system. While other publications report about distinct selected aspects" in the last sentence of Intoduction.
This sentence was so broad without any references. The author should provide what's the difference between your manuscript and previous kinds of literature in detail.
2. The review compares principles of the organization of the brain and the immune system, two important organs developed during 500 million years in multicellular organisms
This sentence in the abstract has no meaning. If you still write down this sentence, you should add this sentence in the introduction, which are based on several references.
3. The table already contains the reference. Please, remove the year of references.
4. Figue 1 was not shown in the revised manuscript. Please, fix it.
5. alpha-lipoic acid is a simple word. Please, don't use abbreviation in this word.
6. 2.1. Immune system seemed to be a mistake.
7. The authors suggested that Table 1 presents some typical features of chapter 2.
However, this table was not matched with the contents.
For examples, Gliotransmitter was reference 2. Gap junctions were reference 1.
Please, check all.
8. What was this review at the bottom (Stem cell hypothesis) of Table 2?
9. Tables 2 and 3 were not matched with the contents in the results. Please, match those serially.
Author Response
.
Reviewer 3 Report
Comments and Suggestions for Authors
I would like to thank the author for the modifications made to improve the flow, clarity, and perspective conveyed through the review. Nevertheless, I believe further revisions are needed to condense and refine information that I find not particularly relevant for this review. For example, the section on the immune system is overly general, containing information already published in other reviews and book chapters. I recommend summarizing this content and focusing specifically on the interaction with the nervous system and the modes of communication between both systems, highlighting the most recent and relevant findings published in recent years.
In addition, the review is very lengthy, with redundant information that makes it tedious to read and detracts from reaching the central point of the article, which I find very interesting but in need of more thorough editing.
Author Response
I would like to thank the author for the modifications made to improve the flow, clarity, and perspective conveyed through the review. Nevertheless, I believe further revisions are needed to condense and refine information that I find not particularly relevant for this review. For example, the section on the immune system is overly general, containing information already published in other reviews and book chapters. I recommend summarizing this content and focusing specifically on the interaction with the nervous system and the modes of communication between both systems, highlighting the most recent and relevant findings published in recent years.
In addition, the review is very lengthy, with redundant information that makes it tedious to read and detracts from reaching the central point of the article, which I find very interesting but in need of more thorough editing.
Response to reviewer 3
I thank the reviewer for his/her comments which have been very helpful again. Please find in the revised version sentences highlighted in blue which should be removed. Sentences highlighted in red are new and should be included.
Point 1: The section on the immune system has been shortened as requested.
Point 2: The focus has been adressed more specifically on the interaction between the nervous and the immune system.
Point 3: The central point of the article formulated in the new hypothesis has been further elaborated.
Many references about the immune system have been removed and replaced by new references concerning the interaction between both systems.
I thank again the reviewer. The comments helped considerably to improve the manuscript.
Sincerely
Volker Schirrmacher
Round 3
Reviewer 3 Report
Comments and Suggestions for Authors
I would like to thank the author for the editing work done; I believe the focus of the revision is now clearer. I have no further comments on the matter.